# Understanding the Clinical Significance of MUC5AC in Biliary Tract Cancers

**DOI:** 10.3390/cancers15020433

**Published:** 2023-01-09

**Authors:** Katherine K. Benson, Ankur Sheel, Shafia Rahman, Ashwini Esnakula, Ashish Manne

**Affiliations:** 1Department of Internal Medicine, The Ohio State University College of Medicine, Columbus, OH 43210, USA; 2Department of Internal Medicine, Division of Medical Oncology at the Arthur G. James Cancer Hospital and Richard J. Solove Research Institute, The Ohio State University Comprehensive Cancer Center, Columbus, OH 43210, USA; 3Department of Pathology, The Ohio State University Wexner Medical Center, Columbus, OH 43210, USA

**Keywords:** biliary tract cancer, cholangiocarcinoma, gallbladder carcinoma, Mucin 5AC, MUC5AC, prognosis, diagnosis, biliary carcinogenesis, intrahepatic, extrahepatic

## Abstract

**Simple Summary:**

Biliary tract cancers (BTC) are tumors arising from cells lining ducts in the liver that facilitate the transport of bile into the gastrointestinal tract. They are often divided into two broad groups, cholangiocarcinoma and gallbladder cancers. They are aggressive tumors with limited treatment options that are often ineffective and have bad outcomes. In current clinical practice, there are no good tests to identify these cancers in early stages and predict their aggressiveness or response to the available treatments. We shed light on the role of a glycoprotein, MUC5AC, in BTC including its potential impact on biliary cancer development. We discuss the clinical evidence surrounding the use of MUC5AC when detected in BTC patients’ tumor tissue and blood and its potential use in managing these cancers.

**Abstract:**

Biliary tract cancers (BTC) arise from biliary epithelium and include cholangiocarcinomas or CCA (including intrahepatic (ICC) and extrahepatic (ECC)) and gallbladder cancers (GBC). They often have poor outcomes owing to limited treatment options, advanced presentations, frequent recurrence, and poor response to available systemic therapy. Mucin 5AC (MUC5AC) is rarely expressed in normal biliary epithelium, but can be upregulated in tissues of benign biliary disease, premalignant conditions (e.g., biliary intraepithelial neoplasia), and BTCs. This mucin’s numerous glycoforms can be divided into less-glycosylated immature and heavily-glycosylated mature forms. Reported MUC5AC tissue expression in BTC varies widely, with some associations based on cancer location (e.g., perihilar vs. peripheral ICC). Study methods were variable regarding cancer subtypes, expression positivity thresholds, and MUC5AC glycoforms. MUC5AC can be detected in serum of BTC patients at high concentrations. The hesitancy in developing MUC5AC into a clinically useful biomarker in BTC management is due to variable evidence on the diagnostic and prognostic value. Concrete conclusions on tissue MUC5AC are difficult, but serum detection might be relevant for diagnosis and is associated with poor prognosis. Future studies are needed to further the understanding of the potential clinical value of MUC5AC in BTC, especially regarding predictive and therapeutic value.

## 1. Introduction

Biliary tract cancers (BTC) are tumors arising from the biliary epithelium [1]. They are often subclassified into gallbladder cancer (GBC) and cholangiocarcinoma (CCA), the latter of which is subdivided into intrahepatic (ICC) and extrahepatic cholangiocarcinoma (ECC); ECC can be further subdivided into perihilar and distal [1,2]. These cancers are relatively uncommon with around 200,000 cases reported worldwide in 2017 [3,4,5]. However, there has been a 76% increase in incidence and 65% increase in mortality globally over recent decades which is very concerning [3]. The 5-year survival rate for metastatic BTC is reported to be around 2–3% [6]. While surgical resection is often curative in early-stage BTC, around 60–70% of cases are diagnosed at a metastatic or advanced stage. Moreover, recurrence rates are high despite using adjuvant therapy (57–87% depending on subtype) in early-stage tumors [7,8]. The gemcitabine/cisplatin combination (GC) has been the standard of care for first-line systemic therapy in advanced BTC (unresectable/locally advanced and metastatic) for many years until the TOPAZ-1 trial recently showed the benefit of adding durvalumab [9,10]. The options for subsequent lines are limited. Therapies directed against mutations in targets such as fibroblast growth factor receptor 2 (*FGFR2*), human epidermal growth factor receptor 2 (HER2), isocitrate dehydrogenase 1 (*IDH1*), neurotrophic tyrosine receptor kinase (NTRK), and *BRAF* are effective, but the prevalence of these mutations in BTC is low [9,10,11,12,13,14,15,16,17,18]. Frequent diagnosis at advanced stages, limited systemic treatments, no screening guidelines, and unreliable biomarkers (diagnostic, prognostic, and predictive) are a few of the main reasons for poor outcomes for BTC patients [19,20,21].

BTCs are diagnosed by tissue biopsy. Clinicians rely on imaging to monitor the treatment response and detect recurrence. Although potential biomarkers have been studied, such as carcinoembryonic antigen (CEA) or carbohydrate antigen 19-9 (CA 19-9), there is variability in prognostic outcomes in the literature and there is no gold standard biomarker in current use [22]. In the recent literature, the reported sensitivity (60–72%) and specificity (75–92%) of serum CA 19-9 for diagnosing CCA or GBC was variable [22,23,24]. Similarly, among studies investigating serum CEA for diagnosis of CCA, the reported sensitivity and specificity ranged from 42 to 85% and 70 to 89%, respectively [25,26,27,28]. Unfortunately, among this available literature, the comparison groups are healthy controls and benign biliary disease; there is a lack of research on the ability of CA 19-9 or CEA to differentiate BTC from non-biliary cancers. This represents an important gap in the literature for diagnostic usefulness in situations where several cancers are in the differential. Regarding prognostic value, the Wang nomogram reported that CA 19-9 and CEA levels were both independent predictors of poor survival in ICC [29,30]. Several other studies also indicated serum CA 19-9 as a potential predictor of poor prognosis, but their recommended cut-off values are highly variable, ranging from 1 to 200 units/mL [31,32,33,34]. Interestingly, another study found that while CEA (at a cut-off of 4.55 µg/L) could predict poor survival of CCA patients in multivariate analysis, CA 19-9 did not [28]. A different study (of ICC patients) determined that the optimal cut-off value for CEA to predict poor prognosis was 9.6 ng/mL [34]. Clearly, there are mixed data in the literature about the usefulness of CA 19-9 or CEA as BTC biomarkers, especially regarding prognostic cut-off values and diagnostic accuracy in undifferentiated cancer. Furthermore, because CA 19-9 is purely a sialyl-Lewis A blood group antigen (Le^a^), it cannot be used as a diagnostic biomarker for cancer in approximately 10% of the population who are Lewis-negative (who cannot physiologically produce CA 19-9) [35]. Additional research is needed to clarify meaningful biomarkers for BTC patients.

One potential biomarker which has been studied in BTC is mucin 5AC (MUC5AC) [36,37,38,39,40,41,42,43,44]. This literature review briefly introduces MUC5AC, its expression in biliary tissues and BTC, its possible role in pathogenesis, diagnostic value, prognostic relevance, and directions for future study.

## 2. Overview of Mucins and MUC5AC

Mucins comprise a family of glycoproteins that are expressed on various epithelial cell surfaces in the body, including respiratory, salivary, gastrointestinal, reproductive, and secretory duct systems [45]. These mucins include tandem repeats containing proline, threonine, and serine, which are rich in post-translational O-linked glycosylation. Variation in tandem repeats, O-linked glycosylation, and other modifications create a wide array of mucin varieties. Mucins are important for the physiologic functioning of many normal epithelial surfaces, and are often modified in tissues based on the specific needs [46]. Mucins have been extensively studied in the gastrointestinal tract, where each organ relies on different mucins for transport, secretion, and protection against bacteria [47]. Human mucins can be divided into two categories: gel-forming mucins which are secreted to form a mucus layer (including MUC2, MUC5AC, MUC5B, and MUC6) and transmembrane mucins which are expressed on the apical surface of epithelial cells (including MUC1, MUC3, MUC4, MUC13, and MUC17) [48]. MUC4 is a ligand for erythroblastic oncogene B2 (ErbB2) receptor tyrosine, and MUC1 affects the immune response to tumor cells and aids in metastasis [37,49,50,51]. Expression of both mucins is often associated with poor outcomes. MUC5B expression was studied in many GB pathologies (stones, infection, and inflammation), but its clinical significance in GBC or BTC has not been determined yet [52,53,54]. MUC2 has an anti-inflammatory effect, and its expression does not seem to affect the outcomes in BTCs [50,55]. MUC6 is a “protective mucin”, a good prognostic marker, and is often detected in well-differentiated tumors [50,56].

Previously, MUC5A, MUC5AB, and MUC5C were considered three different mucins but mapping studies later showed that MUC5A and MUC5C were the same, hence the name, MUC5AC [57]. It has been identified in the normal function of the respiratory epithelium, stomach, salivary glands, reproductive epithelium, and other tissues [58]. MUC5AC is encoded by the *MUC5AC* gene located on chromosome 11 [59]. MUC5AC has also been studied in diseases including malignancies of the lungs, stomach, colon, breast, pancreas, and other organs [60,61,62,63,64,65,66]. Increased expression of MUC5AC has been associated with a poor prognosis in some cancers (including uterine, small bowel, and KRAS-mutant lung cancers), whereas expression of MUC5AC has been potentially linked with an improved prognosis in gastric, ovarian, and colon cancers [67,68,69,70,71,72]. Prognostic impact of MUC5AC in pancreatic cancer is unclear with mixed outcomes [73].

Secreted mucins in humans can undergo O-glycosylation. During this modification, various monosaccharides are attached to immature mucins intracellularly via enzymes through the Golgi and endoplasmic reticulum. Mucins subsequently reach the apical surface and are secreted in mature heavily-glycosylated form [74]. Research in human colorectal adenocarcinoma cell lines has shown that MUC5AC molecules similarly undergo an assembly process in cells to proceed from non-glycosylated immature monomers to glycosylated dimers to fully glycosylated mature oligomers which are secreted [75].

The various forms of MUC5AC (heavily glycosylated vs. less glycosylated) can be detected with different antibodies. CLH2 antibodies against the core tandem repeat region of MUC5AC have been shown to bind weakly to heavily glycosylated mature MUC5AC (and strongly to immature or less glycosylated MUC5AC) in epithelial cytoplasm and are considered antibodies against “immature” MUC5AC [76,77,78,79]. Another antibody, M5P-b1, is also against the core tandem repeat region and stains epithelial cytoplasm similar to CLH2. M5P-b1 will be considered as binding immature MUC5AC in this review [80]. On the other hand, antibodies which bind strongly to both heavily-glycosylated and less-glycosylated MUC5AC are considered antibodies against “mature” MUC5AC. These include MAN-5ACI, Lum5-1 EU-batch, 21M1, and 45M1 [76,81,82]. For some antibodies (such as MSVA-109, polyclonal antibody (Dako, Germany), and manufactured ELISA kits), it is unknown whether they bind immature or mature MUC5AC (see Appendix A in Appendix A for a list of relevant MUC5AC antibodies for this review) [38,83,84].

In the literature, aberrant glycan variants of MUC5AC are associated with adenocarcinomas from various tissues (stomach, ampulla of Vater, colon, lung, breast, and ovary) [66]. In another study, mucinous ovarian cancers are associated with blood-group-ABH glycan variants of MUC5AC compared to control ovaries or serous ovarian cancers (using PLA verified with mass spectrometry) [85]. Aberrant glycosylation of MUC5AC has also been studied in pancreatic cancer (with various techniques including immunohistochemistry, mass spectrometry, and serum ELISA) where certain variants of MUC5AC are associated with malignant transformation and promotion of carcinogenesis [60,61,62,63,64,65].

The remainder of this review will focus on the potential role of MUC5AC in biliary tissues and BTC.

## 3. MUC5AC Expression in Biliary Tissues (Normal, Benign Disease, and Premalignant)

### 3.1. Normal Biliary Tissues

MUC5AC expression in normal biliary tissues, benign biliary diseases, and premalignant lesions is summarized in Table 1. As a note, among biliary neoplasms (benign or malignant), MUC5AC positivity is sometimes used to label neoplasms as “gastric”, “gastric foveolar”, or “pyloric” subtypes because MUC5AC was traditionally seen as a gastric mucin [44,86,87,88,89]. MUC5AC-positive biliary neoplasms are also sometimes described as “classical-type” [90]. On the other hand, “intestinal”, “biliary”, and “null” biliary neoplasms are MUC5AC negative or have low MUC5AC expression [44,86,87,88,91,92].

The literature suggests that MUC5AC is rarely expressed in the normal epithelium of biliary ducts or peribiliary mucous glands but is frequently expressed in normal gallbladder surface epithelium. Importantly, this distinction is independent of whether immature or mature MUC5AC antibodies are used [37,40,80,83,90,93,94,95,96].

### 3.2. Benign Biliary Disease

MUC5AC is expressed or upregulated in many benign biliary disease tissues, as detected using immature CLH2 or M5P-b1 antibodies, mature 45M1 antibodies, or polyclonal Dako antibodies, including primary sclerosing cholangitis (PSC), hepatolithiasis, recurrent pyogenic cholangitis, bile duct adenomas, gallbladder adenomas, and chronic cholecystitis [80,84,90,94,96,97,98,99,100]. There might be a relatively higher expression detected in biliary epithelial cells (bile duct or gallbladder) than peribiliary mucus glands or goblet cells (as seen in cholecystitis with 45M1 antibodies or hepatolithiasis/biliary obstruction tissues with M5P-b1 antibodies) [80,97]. At first glance, MUC5AC upregulation seems to be more consistent in studies on chronic cholecystitis (93–94.3% positivity in three studies using various antibodies) than studies on hepatolithiasis (17%, 40%, and 89% in three studies, respectively, using immature antibodies) [80,84,97,98,99,100]. However, no studies directly compared MUC5AC expression levels between benign obstructive diseases of the bile ducts versus gallbladder to see if there are any statistically significant differences between these two classes. Lastly, the research on serum or bile MUC5AC levels in benign biliary disease (not shown in Table 1) is somewhat limited. One study using mature Lum5-1 EU-batch antibodies via Western Blot found that MUC5AC was present in the bile of patients with PSC and other benign biliary diseases, but much lower in the serum (although statistical comparisons were not drawn) [37]. Another study using USCN Life Science ELISA kit on serum found that serum MUC5AC levels were higher in patients with benign biliary disease than in healthy controls but did not perform statistical comparisons to see if the difference was significant [38].

### 3.3. Premalignant Biliary Diseases

Immature MUC5AC (detected by CLH2 antibodies) is upregulated in many BTC precursor neoplasms such as biliary intraepithelial neoplasia (BilIN), intraductal papillary neoplasm of the bile duct (IPNB), gallbladder dysplasia, biliary dysplasia, and intraductal papillary neoplasm of the liver (IPNL) [44,86,93,94,98,99,100,101]. A few studies show mature MUC5AC (detected with 45M1 antibodies) is upregulated in gallbladder dysplasia, biliary dysplasia, and gallbladder pyloric gland adenoma [44,88,97]. While more research on premalignant lesions using mature MUC5AC antibodies such as 45M1 is needed, the upregulation of MUC5AC as detected with CLH2 immature antibodies across various premalignant lesions suggests possible cellular mechanisms increasing production of MUC5AC in biliary carcinogenesis.

## 4. MUC5AC Expression in BTC Tissue

Studies analyzing MUC5AC expression in BTC tissues via IHC are summarized in Table 2. Among studies which analyzed CCA tissues using immature CLH2 antibodies, the percentage of MUC5AC-positive tumors ranged from 8% to 100% [44,50,83,90,93,99,100,102,103,104,105]. For studies using antibodies against mature MUC5AC (MAN-5ACI, 45M1, or novel S121), the percentage of MUC5AC-positive CCA tumors ranged from 26.5% to 93% [40,44,95,106]. Only one study used both immature (CLH2) and mature (45M1) antibodies on the same tissue (ICC in this case) [44]. The results showed that 34 tumors were stained with both antibodies, 6 were stained with only CLH2, and 6 were stained with only 45M1, revealing no difference in staining frequency between the two antibodies. However, a difference in the staining location was noted (cytoplasm for CLH2 and cytoplasm/biliary lumen/extracellular stroma for 45M1 antibodies). Another study using MSVA-109 antibodies (unknown maturity) found 21% of CCA to be MUC5AC-positive [83]. Interestingly, MUC5AC tissue expression in CCA may vary depending on site. Three studies found that MUC5AC was more frequently expressed in perihilar or hilar (adjacent to the hilum) ICC than peripheral (non-hilar) ICC [44,104,105]. Another study found that ECCs expressed more MUC5AC than ICCs (70.6% vs. 47.1%) [50]. These results suggest that perihilar ICC and ECC may be more likely than peripheral ICC to express or upregulate MUC5AC, perhaps due to an unknown difference in location-specific carcinogenesis mechanisms.

In studies that analyzed GBC using immature CLH2 antibodies, the percentage of MUC5AC-positive tumors ranged from 16.7% to 81.8% [50,94,98,107]. One study that analyzed GBC with mature 45M1 antibodies found 80% of GBC overall to be MUC5AC-positive, and another using polyclonal Dako antibodies (unknown whether immature or mature) observed this in 51.9% of GBC [84,97]. Finally, one study which used mature 21M1 antibodies on general BTC (CCA and GBC tissues) found only 10% of samples to be MUC5AC-positive [37].

Evidently, there is a wide variability in the literature regarding reported tissue-expression levels of MUC5AC in BTC. Although there are ample differences in study methods such as the cancer subtype analyzed, MUC5AC glycoform targeted, or positivity threshold selected, there is no clear association between these factors and the variability of the evidence.

## 5. MUC5AC in Biliary Pathogenesis and Carcinogenesis

The role of MUC5AC in biliary tumorigenesis and metastasis is unclear but we obtained an insight into it through the work of Silsirivanit et al. [42]. They developed a novel monoclonal antibody CA-S27 from pooled CCA tissues that bound to a Lewis-a (Le(a))-associated glycan conjugated to mature MUC5AC [42]. High levels of CA-S27-MUC5AC (i.e., mature MUC5AC) were detected in CCA patients, and had reliable diagnostic and prognostic value (see Table 3 and Table 5). The same experiment showed that suppression of CA-S27-MUC5AC expression in CCA cell lines significantly reduced proliferation, adhesion, migration, and invasion. This study’s results possibly suggest that mature MUC5AC may be involved in pathways in CCA to promote carcinogenesis and metastasis and is similar to its established effect in other tumors [58,108]. The interactions of MUC5AC with other molecules/mutations in biliary pathologies and cancers are summarized in Figure 1 and discussed below.

Aquaporin-1 (AQP-1) has been proposed as a regulator of MUC5AC expression in ICC [109]. AQP-1 is an aquaporin channel that serves as a water transporter for bile in normal cholangiocytes [110]. They are involved in bile secretion and microbial infections. AQP-1 is upregulated in biliary dysplasia but downregulated in invasive ICC. Decreased AQP-1 expression in ICC was associated with increased MUC5AC expression (detected with CLH2 immature antibodies). Additionally, decreased AQP-1 was associated with lymph node metastasis and increased MUC5AC was associated with decreased survival. The authors hypothesized that the downregulation of AQP-1 induces MUC5AC expression in invasive ICC and suggested that AQP-1 may serve a direct role in ICC carcinogenesis. Notably, AQP-1 has been potentially linked with carcinogenesis in other cancers, including lung and colon [111,112].

Research has also connected ecotropic virus integration site 1 protein homolog (EVI-1, a transcription factor) with MUC5AC expression in ICC [113]. EVI-1 upregulation was seen in half of ICC tumors and all IPNBs. EVI-1-positive ICC is a more aggressive disease with advanced stage at diagnosis and decreased survival. EVI-1-positive ICC was more likely to express MUC5AC. An anti-EVI-1 molecule called pyrrole-imidazole polyamide PIP1 was recently designed which inhibits EVI-1 in vitro, and the study authors suggested PIP1 should be investigated as a possible targeted treatment for EVI-1-positive ICC. This could point to a potential role for the measurement of EVI-1 and subsequent EVI-1-targeted treatment in MUC5AC-positive ICC patients.

Trefoil Factor 1 ((TFF1); an estrogen-responsive signaling protein) and MUC5AC were upregulated and correlated in hepatolithiasis, biliary dysplasia, and CCA [114,115]. When TTF1 was applied to CCA cell lines, invasion was stimulated. Additionally, when cell lines were grown in media lacking estrogen agonists, TFF1 expression decreased. The results suggest that TFF1/MUC5AC interactions may be important in the pathogenesis of CCA and suggest a potential role for treatment of TFF1-positive CCA with estrogen antagonists. Further research could investigate a possible utility for measuring TFF1 expression in MUC5AC-positive CCA.

MUC5AC expression in BTC precursor neoplasms was linked with mutations in KRAS (a signaling protein in the RAS/MAPK pathway) and the upregulation of the Wnt/β-catenin pathway [87,116]. Research has also linked the Wnt pathway to upregulation of MUC5AC in cultured biliary cells [117]. Treatment of cells with micro-RNA miR93 to repress the Wnt pathway led to decreased MUC5AC. The RAS/MAPK and Wnt/β-catenin pathways are involved in cell proliferation and homeostasis, and have also been linked to carcinogenesis in many gastrointestinal cancers. Therefore, a potential association with MUC5AC expression in BTC precursor neoplasms and biliary cell lines could point to interactions of MUC5AC with the cellular machinery involved in biliary carcinogenesis [118,119]. Research has also linked mutations in chromatin modifiers (ARID1A, BAP1, and KMT2C) with MUC5AC-positive IPNBs which are similar to mutations seen in CCA, which points to the possible aberrant epigenetic regulation of MUC5AC in BTC [116].

Cholelithiasis is a known risk factor for BTC [120]. A study showed that cholesterol crystals are associated with the upregulation of inflammasomes and increased expression of MUC5AC in gallbladder tissue from cholelithiasis patients [121]. The results indicated that MUC5AC secretion could be decreased in cultured cholesterol-exposed biliary cells through three ways: by inhibiting inflammasomes with siRNAs, inhibiting IL-1 receptor, or inhibiting caspase-1 with Ac-YVAD. These results point to links between cholestasis and MUC5AC expression that could potentially be relevant for using MUC5AC to track inflammation in patients with cholestasis and stratifying their risk for BTC.

Chronic infections and hepatolithiasis are known risk factors for BTC, and these conditions can be associated with chronic exposure of the biliary tract to bacterial lipopolysaccharide (LPS) [122,123,124]. Research on biliary cells has shown that LPS exposure is associated with the upregulation of MUC5AC, possibly through the upregulation of p38-mitogen-activated protein kinase (p38 MAPK) and the downregulation of micro-RNA miR-130b (which disinhibits transcription factor Sp1 which binds the MUC5AC promoter) [125,126,127]. Treatment of LPS-exposed cultured biliary cells with small interference RNA (siRNA) to silence p38 MAPK led to the downregulation of MUC5AC and inflammatory cytokines. Inhibition of cyclo-oxygenase 2 (COX-2) or antagonism of prostaglandin E2 (PGE2) in LPS-treated cultured biliary cells were associated with reduced MUC5AC expression [128]. PGE2 caused upregulation of p38 MAPK, and treatment with a p38 MAPK inhibitor correlated to reduced MUC5AC. MUC5AC and PGE2 were elevated in the bile of hepatolithiasis patients. The literature illuminates the important roles of LPS and p38 MAPK in pathways of biliary inflammation, which correlates with MUC5AC expression.

EGFR (epithelial growth factor receptor) is an important player in normal cell proliferation, but mutations or upregulation in EGFR are associated with many cancers [129]. LPS upregulates EGFR activation in human-cultured biliary epithelial cells [130]. When EGFR activation was inhibited, MUC5AC was downregulated. Treatment of rats affected by chronic proliferative cholangitis ((CPC); a hyperproliferative disorder connected with carcinogenesis) with an anti-EGFR monoclonal antibody (panitumumab) was associated with decreased EGFR expression, lower MUC5AC expression, and decreased hyperproliferation [131]. These results suggest a possible connection between EGFR and MUC5AC expression in biliary cells, and point to a possible role for EGFR-inhibitors in MUC5AC-positive biliary proliferative disorders, although more human research is needed.

Overall, the exact mechanisms of MUC5AC contributing to BTC carcinogenesis are currently unknown. More research is needed to understand it better.

## 6. Does MUC5AC Have Diagnostic Value in BTC?

### 6.1. Tissue MUC5AC Testing

The ambiguous sensitivity of MUC5AC for BTC in tissue samples can be inferred from the numerous BTC tissue studies mentioned earlier, where the percentage of MUC5AC-positive tumors for CCA and GBC was highly variable, ranging from 8% to 100% (CCA) and 16.67% to 90% (GBC), depending on tumor subtype and MUC5AC antibodies (see Table 1). Many studies did not compare the frequency of MUC5AC expression between BTC, other diseases, and/or healthy subjects. However, differences in MUC5AC expression between BTC and other conditions have occasionally been studied in recent years.

Tissue MUC5AC testing, specifically immature MUC5AC (CLH2-reactive), can distinguish malignant tissues (CCA) from healthy controls but not from benign or premalignant designs such as PSC, BilIN, and IPNB [90,93,99]. As mentioned earlier, several studies on gallbladder tissues (with immature CLH2 or unknown-maturity polyclonal Dako antibodies) found significantly lower MUC5AC expression in invasive GBC than gallbladder polyps, adenomas, or chronic cholecystitis [84,94,98]. In one study, mature MUC5AC (21M1-reactive) was detected in just 10% of BTC tissues but not in any healthy biliary tissues [37]. Another study using CLH2 antibodies found that pancreatic ductal carcinoma tissues expressed MUC5AC significantly more frequently than ICC [75]. In addition, a 2004 study using CLH2 antibodies on various carcinoma tissues found that MUC5AC was variably expressed in different gastrointestinal cancers (26% of colorectal, 67% of esophageal, 45% of CCA, 73% of pancreatic ductal, and 55% of stomach carcinomas) but did not draw statistical comparisons between these cancers [76].

Overall, there is obvious variability in detected MUC5AC expression in BTC tissues and unclear usefulness in differentiating BTC from other conditions in tissue samples. However, research is limited with variable study methods, and therefore more studies are needed to elucidate the potential diagnostic relevance of tissue MUC5AC expression for BTC.

### 6.2. Serum MUC5AC Testing

Detection of MUC5AC in serum or bile has been widely studied in recent years for diagnostic purposes in BTC (Table 3). A 2016 systematic review and meta-analysis looked at serum MUC5AC across six studies [36,37,38,40,95,132,133]. This meta-analysis found the pooled area under the curve (AUC) for diagnostic performance of serum MUC5AC for distinguishing BTC from a variety of conditions with statistical significance (including normal subjects, benign biliary diseases, and non-biliary GI cancers) was 0.9138 indicating excellent performance (the range of reported AUC values among the studies was from 0.814 to 0.97). Authors suggested serum MUC5AC could be used to confirm BTC with excellent performance when an indeterminate biliary lesion is found on imaging. However, their data also suggested that serum MUC5AC could miss early CCA. Similar findings were reported in another meta-analysis from 2019 (which included one extra study) [41,134].

Notably, one study also examined bile MUC5AC and noted serum MUC5AC was higher in CCA than benign biliary disease and bile MUC5AC was higher in benign biliary disease [132]. The authors found that a serum/bile MUC5AC ratio demonstrated greater diagnostic performance (AUC 0.97) for CCA than serum MUC5AC alone (AUC 0.82). Finally, three other studies with varying results are described in Table 3 [42,43,135]. Of note, two analyzed serum biomarker panels which showed potential promise for distinguishing CCA from PSC or healthy patients [43,135]. Meanwhile, whilst serum MUC5AC may potentially have relevance as a future diagnostic tool for BTC (either alone or in a panel), there is a variability in performance across the literature, which needs to be examined.

**Table 3 cancers-15-00433-t003:** Studies analyzing the diagnostic performance of serum MUC5AC for BTC.

Study	Tumor Type: (*n* Samples)	Comparison Group(s)	Specimen Source	Lab Technique	MUC5AC Antibody Variant and Dilution Used	Immature or Mature MUC5AC?	Cut-off Threshold for Positivity	Positive vs. Negative MUC5AC Tumors (N)	Sensitivity and Specificity of MUC5AC for BTC	AUC
Wongkham 2003 ^*^ [95]	CCA (various types): 179	Hepato-pancreato-GI cancers: 60; BBD: 62; Active opisthorchiasis: 60; Healthy: 74.	Serum	Immuno-blotting	MAN-5ACI, 1:10,000	Mature	ND	112 vs. 67	62.6%; 96.9% ^A^	ND
Bamrungphon 2007 ^*^ [133]	CCA (various types): 169	BBD: 30; GI cancers: 30; Active opisthorchiasis: 30; Healthy: 30.	Serum	Sandwich ELISA	mAB-22C5, 5 microgm/mL	Mature	OD 0.074	120 vs. 49	71.01%; 90% ^B^	0.8141 (95% CI: 0.763–0.864)
Matull 2008 ^*^ [37]	BTC (Perihilar CCA, ECC, and GBC): 39	PSC: 7; non-biliary malignancy: 5; BBD: 15.	Serum	Western blot	Lum5-1 EU-batch, 1:600	Mature	See below ^C^	17 vs. 22	44% ^D^; 96% ^E^	ND
Silsirivanit 2011 ^*^ [40]	CCA (various types): 97	BBD: 43; non-biliary malignancy: 47; Active opisthorchiasis: 52; healthy: 51.	Serum	Lectin-capture ELISA	S121, 1 microgm/mL	Mature	OD 450 nm	85 vs. 12	87.63%; 89.58% ^F^	0.956 (95% CI: 0.934–0.977)
Danese 2014 ^*^ [132]	CCA (various types, 85% perihilar CCA): 26	BBD (Cholelithiasis: 10; Cholangitis: 10)	Serum	ELISA	ELISA kit from USCN Life Science	Unknown	10.5 ng/mL	ND	80.00%; 73.1% ^G^	0.82 (95% CI: 0.68–0.92)
			Bile	ELISA			6.25 ng/mL	ND	75.00%; 76.9% ^H^	0.85 (95% CI: 0.71–0.93)
			Serum/bile ratio	ELISA			Serum/bile ratio 0.85	ND	92.30%; 95%	0.97 ^K^ (95% CI: 0.87–0.99)
Ruzzenente 2014 * [38]	BTC (Perihilar CCA, ICC, ECC, and GBC): 49	Cholelithiasis: 20; Hepatolithiasis: 3; Healthy: 16.	Serum	ELISA	ELISA kit from USCN Life Science	Unknown	10.5 ng/mL	ND	71%; 94.7% ^L^	0.909
Silsirivanit 2013 [42]	CCA (various types): 96	BBD: 39; Active opisthorchiasis: 52; non-biliary GI malignancy: 48; Healthy: 51.	Serum	Sandwich ELISA	CA-S27, 1 microgm/mL	Mature	OD 0.0268 nm	84 vs. 12	87.5%; 58.8% ^M^	0.822 (*p* < 0.001 for distinguishing CCA from control groups)
Cuenco 2018 [43]	CCA (various types): 66	PSC: 62	Serum	ELISA	ELISA kit from Elabscience	Unknown	0.67 ng/mL	ND	60.6%; 82.3% ^N^	0.72 ^P^ (95% CI: 0.631–0.809)
Kimawaha 2021 [135]	CCA (various types): 40	Non-biliary GI malignancy: 40; Healthy: 40	Serum	Sandwich ELISA	CSB-E10109h ELISA kit from Cusabio	Unknown	104.6 ng/mL (for CCA vs. healthy) ^Q^	ND	52.5% ^R^ 77.5% ^S^	0.639 (95% CI: 0.517–0.762)

MUC5AC = mucin 5AC; BTC = biliary tract cancer; CCA = cholangiocarcinoma; ICC = intrahepatic CCA; ECC = extrahepatic CCA; GI = gastrointestinal; BBD = benign biliary disease; PSC = primary sclerosing cholangitis; GBC = gallbladder adenocarcinoma; AUC = area under the curve; CI = confidence interval; ND = Not described. * = Pooled AUC for these 6 studies in a 2016 meta-analysis was 0.9138; A = Serum MUC5AC expression was more frequent in CCA patients than the comparison groups (*p* < 0.001); B = The mean value of serum MUC5AC in CCA patients was significantly elevated compared to GI cancer, BBD, opisthorchiasis, and healthy patients (*p* < 0.001); C = The strength of Western blot bands was assessed in relation to positive control of sputum (1:10) using a semiquantitative score (negative, +, ++, +++). Band signals of one + strength or greater considered positive; D = Combining serum MUC5AC and bile MUC4 analyses increased sensitivity for detecting BTC to 58% but decreased specificity to 87%; E = Serum MUC5AC levels significantly higher in BTC than BBD (*p* < 0.01); F = The median serum S121 value was elevated significantly in CCA patients compared to comparison groups (*p* < 0.001); G = Serum MUC5AC levels significantly higher in patients with CCA than those with BBD (cholangitis or cholelithiasis) (*p* = 0.0002); H = Bile MUC5AC levels were significantly higher in patients with BBD than CCA (*p* < 0.0001); K = Using serum/bile MUC5AC ratio, the AUC for differentiating CCA from cholangitis was 0.94 (95% CI 0.86–1.00; *p* < 0.0001), between CCA and cholelithiasis was 0.99 (95% CI, 0.98–1.00; *p* < 0.0001), and between cholangitis and cholelithiasis was 0.93 (95% CI, 0.82–1.00; *p* = 0.001); L = Serum MUC5AC levels were greater in patients with BTC compared with BBD (*p* < 0.01) and healthy patients (*p* < 0.01); M = Serum CA-S27 levels of CCA patients were significantly higher than controls (*p* < 0.001); N = Serum MUC5AC levels significantly higher in CCA than PSC (*p* < 0.001); P = A panel combining biomarkers PKM2, CYFRA21.1, MUC5AC, and GGT at 90% specificity gave 81.8% sensitivity and AUC 0.903 for differentiating CCA and PSC; Q = No statistically significant difference in serum MUC5AC between non-biliary GI cancer vs. normal (AUC 0.545 with cut-off of 128 ng/mL; *p* = 0.492) or CCA vs. non-biliary GI cancer (AUC 0.581 with cut-off of 90.51 ng/mL; *p* = 0.213); R = A combined panel with biomarkers S100A9, MUC5AC, angiopoietin-2, and CA19-9 to distinguish CCA vs. healthy gave a sensitivity of 90%, specificity of 95%, and AUC 0.975 (*p* < 0.0001); S = Serum MUC5AC levels were significantly higher in CCA patients than in normal healthy patients (*p* = 0.032).

## 7. Does MUC5AC Have Prognostic Relevance in BTC?

Many studies have evaluated the impact of MUC5AC expression on prognostic outcomes in BTC. Tissue studies utilizing IHC are summarized in Table 4 and serum studies in Table 5. Only statistically significant (*p* < 0.05) associations with prognostic factors are described in detail. Among 18 recent studies which examined prognostic factors and/or survival outcomes, there were 13 tissue analyses, and 7 serum analyses [37,38,39,40,41,42,44,50,84,94,106,107,109,133,135,136,137,138].

### 7.1. Tissue MUC5AC Testing

Four tissue studies associated MUC5AC-positive tumors with decreased overall survival (three using immature CLH2 antibodies and one using both immature CLH2 and mature 45M1) [39,44,109,138]. Four studies found no association between MUC5AC and survival; two used immature CLH2 and two used mature antibodies (MAN-5ACI or 21M1) [37,50,106,107]. In contrast, three associated MUC5AC positivity with improved overall and/or disease-free survival, including one which only showed this association for perihilar ECC and no association for distal ECC [84,136,137]. Two of these analyses used immature CLH2 antibodies and one used polyclonal Dako (unknown MUC5AC maturity). Finally, two studies did not discuss survival outcomes relative to MUC5AC [40,94]. These studies are summarized above in Table 4.

One study using mature MAN-5ACI antibodies associated MUC5AC-positive tumors with more advanced TNM tumor staging and increased frequency of perineural invasion [106]. Two analyses (one with CLH2 and one with CLH2/45M1) found an association with increased frequency of lymph node metastasis [39,44]. One study (using CLH2) found association with higher T category (T2 or greater versus T1) [50]. On the other hand, two studies (one using immature CLH2 and one using unknown maturity polyclonal Dako antibodies) associated MUC5AC positivity with decreased tumor size, including one which only showed this association for perihilar ECC [84,136]. The study using polyclonal Dako also found associations with well-differentiated tumors and lower T category (T1 vs. T4). Six other studies that evaluated prognostic factors found no significant associations with MUC5AC positivity, and finally two studies did not discuss prognostic factors.

In the case reports with tissue MUC5AC analysis (not shown in Table 4), MUC5AC-positive IPNB precursor neoplasms have been associated with DIC/thrombosis as well as local recurrence of tumor after surgery and progression to delayed distant metastasis [139,140]. Another case report of a MUC5AC-negative ICC with local spread to the colon and distant brain metastases, which were surgically removed, noted no recurrence 7 years after the surgeries [141].

Overall, the literature is quite mixed on the prognostic relevance of tissue MUC5AC expression in BTC. There are a wide range of study methods (e.g., positivity thresholds), cancer subtypes, and MUC5AC glycoforms analyzed; however, it is unclear whether these differences could explain any of the variation in reported prognostic relevance. The next subsection will discuss prognostic impact of serum MUC5AC.

### 7.2. Serum MUC5AC Testing

As seen in Table 5, six serum analyses (five using mature MUC5AC antibodies and one using a manufactured ELISA kit) associated increased or positive MUC5AC with decreased survival in BTC [37,38,40,41,42,133]. One (using CSB-E10109h Cusabio of unknown MUC5AC maturity) found no significant association with survival [135]. The latter noted that a combined panel of four biomarkers including MUC5AC was associated with decreased survival. Among six analyses using mature MUC5AC antibodies, associations were also found between increased MUC5AC and more advanced TNM staging (three studies), larger tumor size (one study), and increased frequency of lymph node metastasis (one study) [38,41,133]. The relative consistency of associations across the literature between increased serum MUC5AC, decreased survival, and poor prognostic factors could potentially indicate a prognostic role for serum MUC5AC in BTC. However, data are limited, and future study is necessary to clarify its reliability and usefulness.

## 8. Discussion, Future Directions

MUC5AC might be a potentially useful biomarker for the diagnosis and prognostic outlook of BTC, although more research is needed. While it is rarely expressed in normal biliary epithelial cells, it is expressed or upregulated in benign biliary disease, premalignant precursors, and BTC. MUC5AC is upregulated in the serum of BTC patients, and many studies suggest that serum MUC5AC might be used to differentiate BTC from a variety of other conditions with excellent diagnostic performance (although the data on diagnostic performance of tissue MUC5AC are variable and relatively lacking). Data also suggest that while tissue MUC5AC currently has unclear associations with prognosis, serum MUC5AC levels may be associated with worse prognostic factors and decreased survival in BTC. MUC5AC may also be involved in biliary carcinogenesis (either indirectly or directly) via various biochemical interactions, especially in its mature glycosylated forms.

Given the dire need for effective systemic treatments against BTC, prospective biomarkers such as MUC5AC also need to be evaluated in relation to impact on treatment selection and response. Within the literature there is a lack of studies analyzing the effect of MUC5AC-positivity on BTC treatment; consequently, MUC5AC has no demonstrated predictive value in BTC treatment. However, some research is available that outlines future directions. One study on tumor-associated antigen (TAA) epitopes for stimulating cytotoxic T lymphocyte (CTL) responses in CCA patients showed there were 18 TAA-derived epitopes that were associated with CTL response, including a MUC5AC epitope [142]. Survival was significantly increased in CCA patients who had at least two TAA-specific CTL responses compared to one or no responses. The authors used a set of criteria to determine which epitopes would be potential candidates for epitope therapy to stimulate CTL anti-cancer responses in CCA, and they found that MUC5AC was included on this candidate list. TAA-derived epitopes have been studied in several other types of cancer as potential vaccine therapies to stimulate anti-tumor immune response [1,143,144,145,146]. The results of the CCA study point to a need for further studies on the possible utility of MUC5AC epitope vaccine treatment against BTC. Another question that remains to be answered is the possible role of anti-MUC5AC antibody treatments. Although this has not yet been studied as a treatment for BTC, one study (discussed earlier) provides potential clues for a starting point. The serum levels of MUC5AC detected with novel antibody CA-S27 (against mature MUC5AC) were linked with decreased survival in BTC patients, and CA-S27-MUC5AC is possibly directly involved in BTC carcinogenesis [42]. Given that the in vitro inhibition of CA-S27-MUC5AC with neutralizing antibodies in CCA cell lines led to decreased invasion and growth, this points to a need for further research on the effects of anti-MUC5AC antibodies on CCA cell lines and perhaps eventually for BTC patients.

## 9. Conclusions

In summary, although the literature identifies potentially promising associations for serum MUC5AC in the diagnosis and prognosis of BTC, MUC5AC cannot yet be used in clinical practice as a diagnostic or prognostic marker based on the variability and limitations of the available evidence. The variations in the MUC5AC glycoforms used, study populations (e.g., BTC vs. ICC-only vs. ECC-only vs. CCA vs. GBC) analyzed, other study methods utilized, and outcomes reported among the studies makes it difficult to draw concrete conclusions. Future studies must address these issues, and work on understanding the potential predictive and therapeutic value of MUC5AC in BTC.

## Figures and Tables

**Figure 1 cancers-15-00433-f001:**
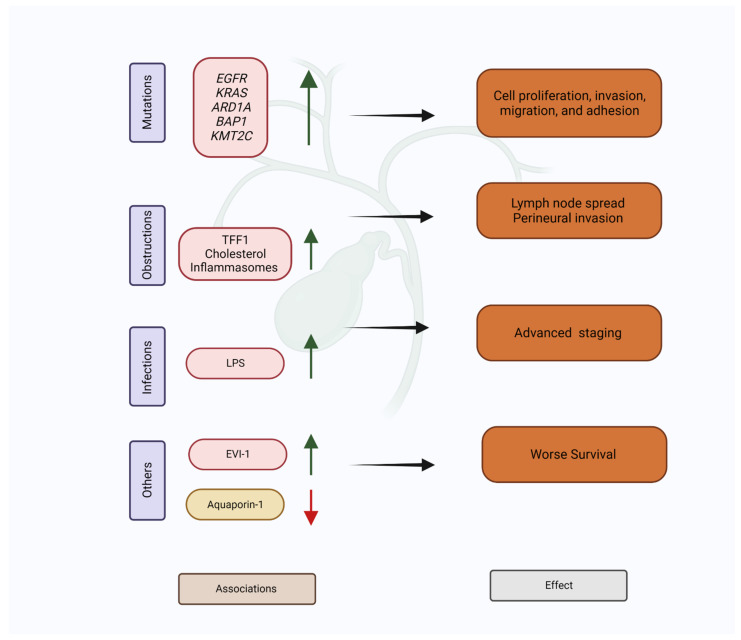
Associations and effects of MUC5AC overexpression in biliary tract cancer. TFF1—Trefoil Factor 1; EVI-1—Ecotropic virus integration site 1 protein homolog; LPS—lipopolysaccharide.

**Table 1 cancers-15-00433-t001:** MUC5AC tissue expression in normal biliary tissues, benign biliary diseases, and premalignant lesions.

	Study	Specimen Source	Percentage of Tissues Positive for MUC5AC	MUC5AC Antibody Variant ^**^	Immature or Mature MUC5AC?	Cut-off Threshold for Positivity
Normal healthy tissue	Rico 2021 [83]	Intrahepatic bile duct epithelium	0%	MSVA-109	Unknown	Weakly positive or greater ^A^
		Gallbladder non-epithelial tissue	0%			
		Gallbladder surface epithelium (columnar cells)	Strong (3+) staining in small fraction of cells			
	Sasaki 1998 [80]	Intrahepatic large bile ducts	4%	M5P-b1	Immature	Not described
		Peribiliary mucous glands	13%			
	Okumura 2020 [93]	Biliary duct epithelium	Some	CLH2	Immature	>10% positive cells
		Peribiliary mucous glands	0%			
	Hughes 2010 [96]	Intrahepatic bile ducts	35.71%	CLH2	Immature	>0% positive cells
		Peribiliary mucous glands	0%			
	Zen 2011 [90]	Intrahepatic bile ducts	0%	CLH2	Immature	>10% positive cells
	Chang 2004 [94]	Gallbladder surface epithelium	91%	CLH2	Immature	≥10% positive cells
	Silsirivanit 2011 [40]	Biliary duct epithelium	Rarely	S121	Mature	>1% positive cells
	Wongkham 2003 [95]	Biliary duct epithelium	0%	MAN-5ACI	Mature	>1% positive cells
	Matull 2008 [37]	Biliary duct epithelium	0%	21M1	Mature	≥20% positive cells
Benign biliary and gallbladder diseases	Sasaki 1999 [97]	Gallbladder surface epithelium in chronic cholecystitis ^B^	93%	45M1	Mature	>1% positive cells
		Gallbladder goblet cells and pseudopyloric glands in chronic cholecystitis ^B^	Rarely			
	Xiong 2012 [83]	Chronic cholecystitis	94.30% ^C^	Polyclonal Dako	Unknown	≥25% positive cells
		Peritumoral gallbladder tissues	82.6%			
		Adenomatous gallbladder polyps	80.0%			
	Bhoge 2017 [98]	Calculous chronic cholecystitis	93.26% ^D^	CLH2	Immature	≥10% positive cells
		Acalculous chronic cholecystitis	73.12%			
		Gallbladder adenomas (3 cases)	100%			
	Sasaki 1998 [80]	Intrahepatic large bile ducts in hepatolithiasis ^E^	89.0%	M5P-b1	Immature	Not described
		Peribiliary mucous glands in hepatolithiasis	22.0%			
		Intrahepatic large bile ducts in EBO	40.0%			
		Peribiliary mucous glands in EBO	16.0%			
	Zen 2006 [99]	Intrahepatic biliary epithelium in hepatolithiasis	40%	CLH2	Immature	>1% positive cells
	Chang 2004 [94]	Gallbladder adenoma	68%	CLH2	Immature	≥10% positive cells
	Zen 2011 [90]	PSC without neoplasia	60% ^F^	CLH2	Immature	>10% positive cells
	Yeh 2005 [100]	Non-neoplastic bile ducts in hepatolithiasis	17%	CLH2	Immature	>10% positive cells
	Hughes 2010 [96]	Bile duct adenomas	90%	CLH2	Immature	>0% positive cells
		Recurrent pyogenic cholangitis	75%			
Premalignant lesions	Okumura 2020 [93]	Low-grade BilIN	88.90% ^G^	CLH2	Immature	>10% positive cells
		High-grade BilIN	93.00%			
	Adsay 2012 [86]	Gallbladder Intracholecystic Papillary-Tubular Neoplasms (ICPN)	55%	CLH2	Immature	≥10% positive cells
	Yeh 2005 [100]	Intraductal papillary neoplasm of the liver (IPNL)	100%	CLH2	Immature	>10% positive cells
	Bhoge 2017 [98]	BilIN (3 cases)	100%	CLH2	Immature	≥10% positive cells
	Sato 2014 [101]	Low-grade BilIN (BilIN 1)	~80%	CLH2	Immature	Mild or greater staining
		High-grade BilIN (BilIN 2 or 3)	100%			
	Zen 2006 [99]	Low-grade BilIN (BilIN 1)	~85% ^H^	CLH2	Immature	≥1% positive cells
		High-grade BilIN (BilIN 2 or 3)	~93%			
		Low-grade IPNB (IPNB1)	~93%			
		High-grade IPNB (IPNB2)	100%			
	Chang 2004 [94]	Gallbladder dysplasia	53%	CLH2	Immature	≥10% positive cells
	Aishima 2006 [44]	Intrahepatic biliary epithelial dysplasia	46.7%	45M1	Mature	>10% positive cells
			40%	CLH2	Immature	
	Albores-Saavedra 2012 [88]	Gallbladder pyloric gland adenomas	95.6%	45M1	Mature	Not described
	Sasaki 1999 [97]	Gallbladder dysplasia	85%	45M1	Mature	>1% positive cells

MUC5AC = mucin 5AC; EBO = extrahepatic biliary obstruction; PSC = primary sclerosing cholangitis; BilIN = biliary intraepithelial neoplasm; BTC = biliary tract cancer; BBD = benign biliary disease; HCS = healthy control subjects; IAC = invasive cholangiocarcinoma; GBC = gallbladder adenocarcinoma; IPNB = intraductal papillary neoplasm of the bile duct. ** = All antibodies noted were applied to tissue specimens via immunohistochemistry (IHC) methods. A = Tumors without staining considered negative. Tumors with 1+ staining intensity in ≤70% of cells or 2+ intensity in ≤30% of cells are considered weakly positive. Tumors with 1+ intensity in >70% of cells, 2+ intensity in 31% to 70%, or 3+ intensity in ≤30% considered moderately positive. Tumors with 2+ intensity in >70% or 3+ intensity in >30% of cells considered strongly positive; B = Among 15 patients with non-dysplastic gallbladder, 10 were chronic cholecystitis with cholelithiasis. The remaining 5 were without cholelithiasis and were removed collaterally and diagnosed as either almost normal gallbladder or mild chronic cholecystitis. For MUC5AC staining, there were no differences between gallbladders with or without cholelithiasis; C = The positive rates of MUC5AC were significantly lower in GBC than those in chronic cholecystitis (*p* < 0.01), peritumoral tissues (*p* < 0.01), or adenomatous polyp (*p* < 0.05); D = Expression rate for MUC5AC significantly lower in acalculous (73.12%) compared to calculous cholecystitis (93.26%) (*p* < 0.001); E = MUC5AC expression in hepatolithiasis large bile ducts was significantly frequent compared with control livers (*p* < 0.01). No significant difference in MUC5AC expression between EBO livers and control livers; F = MUC5AC expression in PSC was significantly higher than in normal bile ducts (*p* < 0.05); G = The number of MUC5AC-positive lesions did not differ significantly between BilIN and IAC (*p* = 0.38 between low- and high-grade BilIN, *p* = 0.40 between high-grade BilIN and IAC, and *p* = 0.62 between low-grade BilIN and IAC); H = MUC5AC expression was more frequently observed in BilINs and IPNBs than non-neoplastic epithelium (*p* < 0.001).

**Table 2 cancers-15-00433-t002:** MUC5AC Expression in BTC Tissues (via IHC analysis).

	Study	Specimen Source	Percentage of Tissues Positive for MUC5AC	MUC5AC Antibody Variant	Immature or Mature MUC5AC?	Cut-off Threshold for Positivity
CCA	Rico 2021 [83]	CCA	21% (Strong 13%, moderate 7%, weak 1%)	MSVA-109	Unknown	Weakly positive or greater ^A^
	Okumura 2020 [93]	CCA	89.1% ^B^	CLH2	Immature	>10% positive cells
	Park 2009 [50]	ICC	47.1%	CLH2	Immature	>10% positive cells
		ECC	70.6% ^C^			
		CCA overall	61.1%			
	Lok 2014 [102]	ICC	12%	CLH2	Immature	≥5% positive cells
	Lau 2004 [103]	CCA	45%	CLH2	Immature	>5% positive cells
	Yeh 2005 [100]	Non-IG-ICC	40% ^D^	CLH2	Immature	>10% positive cells
	Zen 2011 [90]	CCA without PSC	84% ^E^	CLH2	Immature	>10% positive cells
	Zen 2006 [99]	ICC with BIIN	83% ^F^	CLH2	Immature	≥1% positive cells
		ICC with IPNB	100%			
	Guedj 2009 [104]	Hilar ICC	62% ^G^	CLH2	Immature	≥20% positive cells
		Peripheral ICC	22%			
	Akita 2017 [105]	Perihilar ICC ^H^	76% ^K^	CLH2	Immature	>5% positive cells
		Peripheral ICC	8%			
		Hilar CCA	~55%			
	Aishima 2006 [44]	Hilar ICC ^L^	68.8% ^M^	45M1	Mature	>10% positive cells
		Peripheral ICC	26.5%			
		ICC overall	40%			
		Hilar ICC	71.9%	CLH2	Immature	
		Peripheral ICC	25%			
		ICC overall	40%			
	Silsirivanit 2011 [40]	CCA	93%	S121	Mature	Not described
	Boonla 2005 [106]	ICC in patients with history of liver fluke infection	73% (Strong (>25%) expression in 52.8%)	MAN-5ACI	Mature	>1% positive cells
	Wongkham 2003 [95]	CCA	66.6%	MAN-5ACI	Mature	>1% positive cells
GBC	Chang 2004 [94]	GBC	38% ^N^	CLH2	Immature	≥10% positive cells
	Bhoge 2017 [98]	Gallbladder cancers (GBC (15 cases), papillary GBC (2 cases), and carcinosarcoma (1 case))	16.67% ^P^	CLH2	Immature	≥10% positive cells
	Carrasco 2021 [107]	GBC	81.8%	CLH2	Immature	Not described
	Park 2009 [50]	GBC	81.8%	CLH2	Immature	>10% positive cells
	Sasaki 1999 [97]	In situ GBC	90%	45M1	Mature	>1% positive cells
		Invasive GBC	78%			
		GBC overall (in situ and invasive)	80%			
	Xiong 2012 [83]	GBC	51.9% ^Q^	Polyclonal Dako	Unknown	≥25% positive cells
BTC	Matull 2008 [37]	BTC (78.3% CCA, 21.7% GBC)	10%	21M1	Mature	≥20% positive cells

MUC5AC = mucin 5AC; IHC = immunohistochemistry; BTC = biliary tract cancer; CCA = cholangiocarcinoma; ICC = intrahepatic cholangiocarcinoma; ECC = extrahepatic cholangiocarcinoma; non-IG-ICC = non-intraductal-growth-type ICC (which includes mass-forming or periductal-infiltrating ICC types); PSC = primary sclerosing cholangitis; IPNB = intraductal papillary neoplasm of the bile duct; GBC = gallbladder adenocarcinoma; BilIN = biliary intraepithelial neoplasm. A = Tumors without staining are considered negative. Tumors with 1+ staining intensity in ≤70% of cells or 2+ intensity in ≤30% of cells considered weakly positive. Tumors with 1+ intensity in >70% of cells, 2+ intensity in 31%–70%, or 3+ intensity in ≤30% are considered moderately positive. Tumors with 2+ intensity in >70% or 3+ intensity in >30% of cells considered strongly positive; B = The number of MUC5AC-positive lesions did not differ significantly between BilIN and CCA (*p* = 0.38 between low-grade and high-grade BilIN, *p* = 0.40 between high-grade BilIN and CCA, and *p* = 0.62 between low-grade BilIN and CCA); C = A higher degree of MUC5AC expression was observed in ECCs than ICCs (*p* = 0.026); D = MUC5AC was strongly expressed in all IPNLs and was expressed in 75% of well-differentiated and 33% of moderately-differentiated non-IG-ICCs. No poorly differentiated non-IG-ICCs exhibited MUC5AC expression; E = MUC5AC expression was significantly higher in CCA than normal bile ducts (*p* < 0.05); F = MUC5AC expression is more frequently observed in ICC with BilIN (83%) and ICC with IPNB than non-neoplastic epithelium (*p* < 0.001); G = Hilar ICC more often expressed MUC5AC than peripheral ICC (*p* < 0.0001); H = Hilar CCA defined as predominantly involving the right and left hepatic ducts and junction, whereas more proximal tumors were considered ICC. Perihilar ICC is defined as a ductal morphology and minor tubular components (if present) only at the tumor–liver interface. ICC beyond these criteria is called peripheral ICC; K = MUC5AC expressed more frequently in hilar CCA and perihilar ICC than peripheral ICC (*p* < 0.05); L = Hilar ICC classified as tumors as involving the second branch of the bile duct localized in the hilar portion of the liver. Peripheral ICC classified as tumors in the hepatic periphery; M = MUC5AC expression more frequent in hilar than peripheral ICC (*p* < 0.0001) when visualized with 45M1 or CLH2 antibodies; N = MUC5AC expression significantly lower in GBC than normal gallbladder mucosa or gallbladder adenomas (*p* < 0.01); P = MUC5AC expression significantly less frequent in GBC (18 GBC + 3 BilIN) (28.57%) than chronic cholecystitis (87.19%) (*p* < 0.001); Q = MUC5AC expression rates significantly lower in GBC than in chronic cholecystitis (*p* < 0.01), peritumoral tissues (*p* < 0.01), or adenomatous polyp (*p* < 0.05).

**Table 4 cancers-15-00433-t004:** Studies evaluating the prognostic impact of MUC5AC expression in BTC (tissue IHC).

Study	Type of Tumor	MUC5AC Antibody Variant and Dilution	Immature or Mature MUC5AC?	Expression Site(s)	Positivity Expression Threshold	Positive/High vs. Negative/Low MUC5AC Tumors (N)	Prognostic Factor(s) ^A^ Related to MUC5AC Positivity or Level (Positive/High vs. Negative/Low)	Survival Outcome(s) ^B^ Related to MUC5AC Positivity or Level (Positive/High vs. Negative/Low)
Boonla 2005 [106]	ICC with history of liver fluke infection	MAN-5ACI, 1:1000	Mature	Cytoplasm and luminal mucin	>25% positive cells	46 vs. 41	More advanced TNM staging (*p* = 0.008). Increased frequency of neural invasion (*p* = 0.022)	Not statistically significant for OS
Matull 2008 [37]	BTC (ICC, ECC, perihilar CCA, and GBC)	21M1, 1:1,000,000	Mature	Not described	≥20% positive cells	7 vs. 62	Not statistically significant for prognostic factors.	Not statistically significant for OS
Silsirivanit 2011 [40]	CCA (various types)	S121, 5 microgm/mL	Mature	Cytoplasm, apical surface	Not described	42 vs. 3	Not statistically significant for prognostic factors	Survival outcomes and MUC5AC not analyzed with tissue IHC
Aishima 2006 [44]	ICC	45M1, 1:100	Mature	Cytoplasm, luminal surface, and EC stroma	>10% positive cells	26 vs. 14 (for LN metastasis analysis)	Increased frequency of LN metastasis (*p* < 0.0001)	Decreased OS for MUC5AC+ only phenotype compared to MUC6+ only, MUC5AC+/MUC6+, or MUC5AC-/MUC6- (MVA *p* = 0.0042). Specific survival % not reported.
		CLH2, 1:100	Immature	Cytoplasm		24 vs. 16 (for LN metastasis analysis)		
Abe 2015 [39]	ICC	CLH2, 1:100	Immature	Cytoplasm	≥5% positive cells	13 vs. 29	Increased frequency of LN metastasis following curative surgery (*p* = 0.021)	Decreased OS (MVA *p* = 0.005). 3-year OS 13.8% vs. 72.1%
Aishima 2007 [109]	ICC	CLH2, 1:100	Immature	Cytoplasm	>0% positive cells	40 vs. 72	Not discussed	Decreased OS (UVA *p* = 0.0023; MVA *p* = 0.4721). 5-year OS 12.1% vs. 47.8%.
Iguchi 2009 [138]	ICC	CLH2, 1:100	Immature	Cytoplasm	>0% positive cells	25 vs. 36	Not discussed	Decreased OS (UVA *p* < 0.001). 1-year OS 50.4% vs. 75%. 3-year OS 16.8% vs. 51.5%.
Park 2009 [50]	ICC	CLH2, 1:100	Immature	Cytoplasm	>10% positive cells	16 vs. 18	Higher T category (*p* = 0.034 for T2 or greater vs. T1)	Not statistically significant for OS
	ECC					36 vs. 15	Not statistically significant for T category.	
Carrasco 2021 [107]	GBC	CLH2	Immature	Cytoplasm	Not described	135 vs. 30	Not statistically significant for prognostic factors	Not statistically significant for OS
Ishida 2019 [136]	Perihilar ECC ^C^	CLH2, 1:100	Immature	Cytoplasm	Low vs. High (based on hierarchal clustering)	17 vs. 13	Decreased tumor size (*p* = 0.001).	Increased OS (MVA *p* = 0.024). 5-year OS 60.8% vs. 13.9%.
	Distal ECC					29 vs. 25	Not statistically significant for prognostic factors	Not statistically significant for OS
Chang 2004 [94,66]	GBC	CLH2, 1:100	Immature	Cytoplasm	≥10% positive cells	50 vs. 81	Not statistically significant for prognostic factors	Not discussed
Xiong 2012 [83]	GBC	Polyclonal Dako	Unknown	Cytoplasm, cell membrane	≥25% positive cells	56 vs. 52 for prognostic factors; 38 vs. 29 for survival outcomes	Decreased tumor size (*p* < 0.05 for size <2 cm vs. ≥2 cm). Greater % of well-differentiated tumors (*p* < 0.01 for well vs. poorly differentiated). Lower T category (*p* < 0.05 for T1 vs. T4).	Increased OS (MVA *p* = 0.011). Avg OS 11.7 months vs. 9.0 months.

BTC = biliary tract cancer; MUC5AC = mucin 5AC; UVA = univariate analysis; MVA = multivariate analysis; CCA = cholangiocarcinoma; ICC = intrahepatic CCA; ECC = extrahepatic CCA; GBC = gallbladder adenocarcinoma; BilIN = biliary intraepithelial neoplasm; IPNB = intraductal papillary neoplasm of the bile duct; LN = lymph node; OS = overall survival; DFS = disease-free survival. A = Only statistically significant (generally *p* < 0.05) prognostic factors are reported in this table, followed by the *p* value; B = For statistically significant survival outcomes, *p*-values from multivariate analysis (MVA) are reported when available. If this was not available, then univariate analysis (UVA) was reported. If there is a discrepancy in statistical significance between univariate and multivariate analysis, then both are reported; C = Each ECC tumor was classified as perihilar or distal based on the surgical and pathological findings in accordance with the TNM classification.

**Table 5 cancers-15-00433-t005:** Studies evaluating the prognostic impact of MUC5AC expression in BTC (serum).

Study	Type of Tumor	Lab Technique	MUC5AC Antibody Variant and Dilution	Immature or Mature MUC5AC?	Positivity Expression Threshold	Positive/High vs. Negative/Low MUC5AC Tumors (N)	Prognostic factor(s) ^A^ Related to MUC5AC Positivity or Level (Positive/High vs. Negative/Low)	Survival Outcome(s) ^B^ Related to MUC5AC Positivity or Level (Positive/High vs. Negative/Low)
Silsirivanit 2013 [42]	CCA (various types)	Sandwich ELISA	CA-S27, 1 microgm/mL	Mature	OD450 nm = 0.206	32 vs. 64	Not discussed	Decreased OS (UVA *p* < 0.001). Median survival 145 days vs. 256 days.
Boonla 2003 [41]	CCA (various types)	Gel electrophoresis and immunoblotting	MAN-5ACI, 1:10,000	Mature	Any reactivity ^C^	112 vs. 67	More advanced TNM staging (UVA *p* = 0.009 for Stage IVb disease vs. lower stages). Increased tumor size (UVA *p* = 0.01 for tumor sizes >5 cm vs. ≤5 cm).	Decreased cumulative survival (MVA *p* < 0.001). Median survival 127 days vs. 329 days. (Among patients with Stage IVb tumors specifically, *p* < 0.0003 with median survival 116 days vs. 329 days.)
Matull 2008 [37]	BTC (ICC, ECC, perihilar CCA, and GBC)	Western blot	Lum5-1 EU-batch, 1:600	Mature	See below ^D^	17 vs. 22	Not discussed	Decreased cumulative survival (UVA *p* = 0.03). Median survival 5.2 months vs. 16.9 months.
Silsirivanit 2011 [40]	CCA (various types)	Lectin-capture ELISA	S121, 1 microgm/mL	Mature	OD > 0.23 nm	Not described	Not discussed	Decreased survival (UVA *p* = 0.024). Median survival 148 +/− 52 days vs. 224+/− 21 days.
Bamrungphon 2007 [133]	CCA (various types)	Sandwich ELISA	mAb-22C5, 5 microgm/mL	Mature	OD > 0.074 nm	108 vs.44	More advanced TNM staging (UVA *p* < 0.016).	Decreased cumulative survival (UVA *p* < 0.0246). Median survival 158 days vs. 297 days. (CCA patients who survived during the follow-up period also had a significantly lower level of serum MUC5AC than those who died. *p* = 0.026).
Ruzzenente 2014 [38]	BTC (Perihilar CCA, ICC, ECC, and GBC)	ELISA	ELISA kit from USCN Life Science	Unknown	≥14 ng/mL	Not described	Increased frequency of LN metastasis (*p* = 0.05). More advanced TNM staging (*p* = 0.047 for Stage IVb vs. lower stages).	Decreased OS (UVA *p* = 0.039). 1-year OS 51.7% vs. 88.9%. 3-year OS 21.5% vs. 59.3%.
Kimawaha 2021 [135]	CCA (various types)	Sandwich ELISA	CSB-E10109h; Cusabio	Unknown	Not described	Not described	Not statistically significant for prognostic factors.	Not statistically significant for OS. (Although a combined panel of 5 biomarkers including MUC5AC could predict decreased survival. *p* = 0.018 for decreased OS in patients with 4/5 or 5/5 positive biomarkers versus 3/5 or less).

MUC5AC = mucin 5AC; BTC = biliary tract cancer; CCA = cholangiocarcinoma; ICC = intrahepatic CCA; ECC = extrahepatic CCA; GBC = gallbladder adenocarcinoma; LN = lymph node; OS = overall survival; DFS = disease-free survival; UVA = univariate analysis; MVA = multivariate analysis; A = Only statistically significant (generally *p* < 0.05) prognostic factors are reported in this table, followed by the *p*-value; B = For statistically significant survival outcomes, *p*-values from multivariate analysis (MVA) are reported when available. If this was not available, then univariate analysis (UVA) was reported. If there is a discrepancy in statistical significance between univariate and multivariate analysis, both are reported; C = The samples that yielded immunoreactivity on the film were interpreted as positive for MUC5AC; D = The strength of Western blot bands was assessed in relation to positive control of sputum (1:10) using a semiquantitative score (negative, +, ++, +++). Band signals of one + strength or greater were considered positive.

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
