# Peer review of "Understanding the Clinical Significance of MUC5AC in Biliary Tract Cancers"

_cancers, 2023, doi:10.3390/cancers15020433_

Round 1

Reviewer 1 Report

The authors, Katherine et al. review on "Understanding the clinical significance of MUC5AC in Biliary Tract Cancers" is very informative and carries weightage in discussing all the available evidences of MUC5AC expression and detection in various pathological and non-pathological aspects of BTC. Over all the review article is well organized and discussed thoroughly with tables and schematic representations. As the authors discussed in their discussion section about the MUC5AC importance in BTC and more evidence is needed from the future research to show strong correlation, their review article can help in planning and executing the future researchers work in this field.  

Whereas, I would recommend the authors to add additional information on various Mucins and their isoforms in different clinic pathological situations more elaborately and ask to discuss more about key mucins that are highly cited in various cancers like MUC4 etc…  Also, please add MUC5B and its importance and what it causes more different than MUC5AC and why is MUC5AC is very important to study further in BTC excluding others. This will clearly provide a Segway to the main content of this review if they discuss these recommendations as a second paragraph in section-2 “overview of mucins and MUC5AC” before elaborating MUC5AC.

With these recommended changes, I think it has a merit in accepting this review article for the publication in Cancers journal. 

Author Response

Reply to reviewer 1

The authors, Katherine et al. review on "Understanding the clinical significance of MUC5AC in Biliary Tract Cancers" is very informative and carries weightage in discussing all the available evidences of MUC5AC expression and detection in various pathological and non-pathological aspects of BTC. Over all the review article is well organized and discussed thoroughly with tables and schematic representations. As the authors discussed in their discussion section about the MUC5AC importance in BTC and more evidence is needed from the future research to show strong correlation, their review article can help in planning and executing the future researchers work in this field.  

Whereas, I would recommend the authors to add additional information on various Mucins and their isoforms in different clinic pathological situations more elaborately and ask to discuss more about key mucins that are highly cited in various cancers like MUC4 etc…  Also, please add MUC5B and its importance and what it causes more different than MUC5AC and why is MUC5AC is very important to study further in BTC excluding others. This will clearly provide a Segway to the main content of this review if they discuss these recommendations as a second paragraph in section-2 “overview of mucins and MUC5AC” before elaborating MUC5AC.

  • We appreciate the feedback. As the focus was on MUC5AC only, we mentioned other mucins. The following sentences were added to section 2 (lines 107-116)- MUC4 is a ligand for erythroblastic oncogene B2 (ErbB2) receptor tyrosine, and MUC1 affects the immune response to tumor cells and aids in metastasis [1-4]. Expression of both mucins is often associated with poor outcomes. MUC5B expression was studied in many GB pathologies (stones, infection, and inflammation), but its clinical significance in GBC or BTC has not been determined yet [5-7]. MUC2 has an anti-inflammatory effect, and its expression does not seem to affect the outcomes in BTCs [3,8]. MUC6 is a ‘protective mucin’, is a good prognostic marker, and is often detected in well-differentiated tumors [3,9]. Previously, MUC5A, MUC5AB, and MUC5C were considered three different mucins but mapping studies later showed that MUC5A and MUC5C were the same, hence the name, MUC5AC [10].

Reviewer 2 Report

Title: Understanding the Clinical Significance of MUC5AC in Biliary Tract Cancers

This review article highlights the clinical significance of glycoprotein MUC5AC in biliary tract cancers. Cholangiocarcinoma and gallbladder cancers are epithelial cell cancer with an aggressive phenotype. The author has well written the importance of MUC5AC as a diagnostic, therapeutic, and prognostic marker.

Comments

1. The author has explained and highlighted MUC5AC solely in biliary tract cancer. The author can provide the expression and the levels of other mucins (gel-forming and transmembrane mucins) to highlight the superiority of MUC5AC.

2. The author has discussed that variation in the glycosylation of MUC5AC is associated with cancer progression. It would be great if the author could tabulate those variants and their associated biliary tract cancer.

3. On page number 3, line number 134 & 135 needs to be reframed for easy understanding.

4. Table 1 represents the increases in the levels of MUC5AC in the various types of biliary tract cancer; however, non-neoplastic diseases such as cholecystitis and hepatolithiasis also demonstrate higher levels of MUC5AC. Hence, is it accurate to say/suggest that MUC5AC is a diagnostic marker for biliary tract cancer? In addition, it is difficult to understand the figure legends stating intensity and the percent cells (145 to 148). The author should explain this more clearly for easy understanding of the reader, and the abbreviations can be moved to a typical section, “abbreviations” to shorten the table legends.

5. The author needs to highlight the current diagnostic (invasive and non-invasive) strategies available for biliary tract cancer and its drawbacks suggesting MUC5AC could be a potential diagnostic marker.

6. In table 3, the representation of MUC5AC would be more appropriate than mentioning OD values 

Author Response

Reply to reviewer 2

This review article highlights the clinical significance of glycoprotein MUC5AC in biliary tract cancers. Cholangiocarcinoma and gallbladder cancers are epithelial cell cancer with an aggressive phenotype. The author has well written the importance of MUC5AC as a diagnostic, therapeutic, and prognostic marker.

Comments

  1. The author has explained and highlighted MUC5AC solely in biliary tract cancer. The author can provide the expression and the levels of other mucins (gel-forming and transmembrane mucins) to highlight the superiority of MUC5AC.

    • We appreciate the feedback. As the focus was on MUC5AC only, we mentioned other mucins. The following sentences were added to section 2 (lines 107-116, this was also in response to Reviewer 1’s feedback) - MUC4 is a ligand for erythroblastic oncogene B2 (ErbB2) receptor tyrosine, and MUC1 affects the immune response to tumor cells and aids in metastasis [1-4]. Expression of both mucins is often associated with poor outcomes. MUC5B expression was studied in many GB pathologies (stones, infection, and inflammation), but its clinical significance in GBC or BTC has not been determined yet [5-7]. MUC2 has an anti-inflammatory effect, and its expression does not seem to affect the outcomes in BTCs [3,8]. MUC6 is a ‘protective mucin’, is a good prognostic marker, and is often detected in well-differentiated tumors [3,9]. Previously, MUC5A, MUC5AB, and MUC5C were considered three different mucins but mapping studies later showed that MUC5A and MUC5C were the same, hence the name, MUC5AC [10].
  1. The author has discussed that variation in the glycosylation of MUC5AC is associated with cancer progression. It would be great if the author could tabulate those variants and their associated biliary tract cancer.
    • We appreciate the feedback. Based on their glycosylation level, we used an established broad classification of MUC5AC variants into two major kinds, mature and immature type. Based on the available studies, we discussed them extensively in sections 2, 3, 4, 6, and 7. We felt a separate table would be redundant, especially, when we designed tables 1-5 with separate columns to indicate mature vs immature MUC5AC.

  1. On page number 3, line number 134 & 135 needs to be reframed for easy understanding.

    • We appreciate the feedback. We simplified the line as follows, On the other hand, “intestinal”, “biliary”, and “null” biliary neoplasms are MUC5AC negative or have low MUC5AC expression.

  1. Table 1 represents the increases in the levels of MUC5AC in the various types of biliary tract cancer; however, non-neoplastic diseases such as cholecystitis and hepatolithiasis also demonstrate higher levels of MUC5AC. Hence, is it accurate to say/suggest that MUC5AC is a diagnostic marker for biliary tract cancer? In addition, it is difficult to understand the figure legends stating intensity and the percent cells (145 to 148). The author should explain this more clearly for easy understanding of the reader, and the abbreviations can be moved to a typical section, “abbreviations” to shorten the table legends.

    • We appreciate the feedback. In lines 374-378 under section 6.1 and 398-400 under section 6.2, we clarified that there is not enough evidence to use MUC5AC as a diagnostic marker. We asked whether it is a diagnostic marker and answered without bias based on the available evidence.
    • We appreciate the feedback on the table legends. That information is critical to keep the evidence in the right context. We adopted this approach based on our previous publication experience. Putting it in the text would confuse the readers.
    • We appreciate the feedback on the abbreviations. We followed the publisher’s guidelines. If they allow us to have a separate section for it, we are happy to incorporate all of them into one section.

  1. The author needs to highlight the current diagnostic (invasive and non-invasive) strategies available for biliary tract cancer and its drawbacks suggesting MUC5AC could be a potential diagnostic marker.

    • We appreciate the feedback. Addressing this, we rearranged the lines in the introduction and added the following lines - BTCs are diagnosed by tissue biopsy. Clinicians rely on imaging to monitor the treatment response and detect recurrence. Although potential biomarkers have been studied, such as carcinoembryonic antigen (CEA) or carbohydrate antigen 19-9 (CA 19-9), there is variability in prognostic outcomes in the literature and there is no gold standard biomarker in current use [23]. In recent literature, the reported sensitivity (60-72%) and specificity (75-92%) of serum CA 19-9 for diagnosing CCA or GBC was variable [23-25]. Similarly, among studies investigating serum CEA for diagnosis of CCA, the reported sensitivity and specificity ranged from 42-85% and 70-89% respectively [26-29]. Unfortunately, among this available literature, the comparison groups are healthy controls and benign biliary disease; there is a lack of research on the ability of CA 19-9 or CEA to differentiate BTC from non-biliary cancers. This represents an important gap in the literature for diagnostic usefulness in situations where several cancers are in the differential. Regarding prognostic value, the Wang nomogram reported that CA 19-9 and CEA levels were both independent predictors of poor survival in ICC [30,31]. Several other studies also indicated serum CA 19-9 as a potential predictor of poor prognosis, but their recommended cut-off values are highly variable, ranging from 1-200 units/ml [32-35]. Interestingly, another study found that while CEA (at a cut-off of 4.55 µg/l) could predict poor survival of CCA patients in multivariate analysis, CA 19-9 did not [29]. A different study (of ICC patients) determined that the optimal cut-off value for CEA to predict poor prognosis was 9.6 ng/mL [35].  Clearly, there is mixed data in the literature about the usefulness of CA 19-9 or CEA as BTC biomarkers, especially regarding prognostic cut-off values and diagnostic accuracy in undifferentiated cancer. Furthermore, because CA 19-9 is purely a sialyl-LewisA blood group antigen (Lea), it cannot be used as a diagnostic biomarker for cancer in approximately 10% of the population which are Lewis-negative (who cannot physiologically produce CA 19-9) [36].  Additional research is needed to clarify meaningful biomarkers for BTC patients.

  1. In table 3, the representation of MUC5AC would be more appropriate than mentioning OD values

    • We appreciate the feedback. We used OD values only in available studies. The requested information is not clear to us.

Reviewer 3 Report

This is a well-written and comprehensive review on the potential diagnostic and prognostic value of MUC5AC in biliary tract cancer. The following additional information is suggested for minor improvements:

1. Please benchmark MUC5AC over the conventional markers used for biliary tract? One need to show the superiority over the conventional methods to justify its further development. 

2. It seems that MUC5AC is upregulated in a variety of non-cancerous biliary conditions. Does the cohorts described in this study included only healthy controls and biliary tract cancer, without other non-cancer conditions? If any comparison data is available, please describe in greater detail.

3. What is the potential mechanism of MUC5AC in contributing to biliary tumorigenesis and metastasis? This information is lacking in detail in the current review. 

Author Response

Reply to reviewer 3

This is a well-written and comprehensive review on the potential diagnostic and prognostic value of MUC5AC in biliary tract cancer. The following additional information is suggested for minor improvements: 

  1. Please benchmark MUC5AC over the conventional markers used for biliary tract? One need to show the superiority over the conventional methods to justify its further development.  

    • We appreciate the feedback. Addressing this, we added following lines in the introduction (this was also in response to Reviewer 2’s feedback) - BTCs are diagnosed by tissue biopsy. Clinicians rely on imaging to monitor the treatment response and detect recurrence. Although potential biomarkers have been studied, such as carcinoembryonic antigen (CEA) or carbohydrate antigen 19-9 (CA 19-9), there is variability in prognostic outcomes in the literature and there is no gold standard biomarker in current use [23]. In recent literature, the reported sensitivity (60-72%) and specificity (75-92%) of serum CA 19-9 for diagnosing CCA or GBC was variable [23-25]. Similarly, among studies investigating serum CEA for diagnosis of CCA, the reported sensitivity and specificity ranged from 42-85% and 70-89% respectively [26-29]. Unfortunately, among this available literature, the comparison groups are healthy controls and benign biliary disease; there is a lack of research on the ability of CA 19-9 or CEA to differentiate BTC from non-biliary cancers. This represents an important gap in the literature for diagnostic usefulness in situations where several cancers are in the differential. Regarding prognostic value, the Wang nomogram reported that CA 19-9 and CEA levels were both independent predictors of poor survival in ICC [30,31]. Several other studies also indicated serum CA 19-9 as a potential predictor of poor prognosis, but their recommended cut-off values are highly variable, ranging from 1-200 units/ml [32-35]. Interestingly, an-other study found that while CEA (at a cut-off of 4.55 µg/l) could predict poor survival of CCA patients in multivariate analysis, CA 19-9 did not [29]. A different study (of ICC patients) determined that the optimal cut-off value for CEA to predict poor prognosis was 9.6 ng/mL [35].  Clearly, there is mixed data in the literature about the usefulness of CA 19-9 or CEA as BTC biomarkers, especially regarding prognostic cut-off values and diagnostic accuracy in undifferentiated cancer. Furthermore, because CA 19-9 is purely a sialyl-LewisA blood group antigen (Lea), it cannot be used as a diagnostic biomarker for cancer in approximately 10% of the population which are Lewis-negative (who cannot physiologically produce CA 19-9) [36].  Additional research is needed to clarify meaningful biomarkers for BTC patients.

  1. It seems that MUC5AC is upregulated in a variety of non-cancerous biliary conditions. Does the cohorts described in this study included only healthy controls and biliary tract cancer, without other non-cancer conditions? If any comparison data is available, please describe in greater detail. 

    • We appreciate your feedback. In two sections, we have already highlighted various comparisons with non-cancerous biliary conditions.

      • In subsection 6.1 (Diagnostic relevance of tissue MUC5AC) starting on line 360, we state “Tissue MUC5AC testing, specifically immature MUC5AC (CLH2-reactive), can distinguish malignant tissues (CCA) from healthy controls but not from benign or premalignant designs such as PSC, biliary intraepithelial neoplasia (BilIN) and intraductal papillary neoplasm of the bile duct (IPNB)… As mentioned earlier, several studies on gallbladder tissues (with immature CLH2 or unknown-maturity polyclonal Dako antibodies) found significantly lower MUC5AC expression in invasive GBC than gallbladder polyps, adenomas, or chronic cholecystitis”. We emphasize the usefulness of tissue MUC5AC for differentiating BTC from other conditions is unclear. 

      • Next, in subsection 6.2 (Diagnostic relevance of serum MUC5AC) starting on line 381 and Table 3, we mention several comparisons of serum MUC5AC levels between BTC, healthy controls, and/or benign biliary conditions. For example, see the column “Comparison Group(s)” in Table 3 for complete listings of comparison cohorts. Also, please see the legend under Table 3 for important notes on diagnostic relevance of serum or bile MUC5AC for differentiating BTC from benign biliary conditions (legend notes A, B, E, F, G, H, K, L, M, N, and P).

  1. What is the potential mechanism of MUC5AC in contributing to biliary tumorigenesis and metastasis? This information is lacking in detail in the current review.  

    • Thank you for the thoughtful question. Upon our review of the literature, there is limited evidence in the literature on exact mechanisms of MUC5AC contributing to BTC carcinogenesis. We re-arranged the text in section 5 streamlining the evidence.
      • Now, we start with lines, The MUC5AC’s role in biliary tumorigenesis and metastasis is unclear but we got an insight into it through Silsirivanit et al ’s work [43]. They developed a novel monoclonal antibody CA-S27 from pooled CCA tissues that bind to a Lewis-a (Le(a)) associated glycan conjugated to mature MUC5AC [43]. High levels of CA-S27-MUC5AC (i.e., mature MUC5AC) were detected in CCA patients, and had reliable diagnostic and prognostic value (see Tables 3 and 5). The same experiment showed that suppression of CAS-S27-MUC5AC expression in CCA cell lines significantly reduced proliferation, adhesion, migration, and invasion. This study’s results possibly suggest that mature MUC5AC may be involved in pathways in CCA to promote carcinogenesis and metastasis and is similar to its established effect in other tumors [60,110]. The interactions of MUC5AC with other molecules/mutations in biliary pathologies and cancers are summarized in Figure 1 and discussed below.

      • Modified the ending of the section as follows, Overall, the exact mechanisms of MUC5AC contributing to BTC carcinogenesis are currently unknown. More research is needed to understand it better.